# Hyperspectral Image Classification Based on Dense Pyramidal Convolution and Multi-Feature Fusion

**Junsan Zhang** [1,2], **Li Zhao** [1], **Hongzhao Jiang** [3], **Shigen Shen** [4,*], **Jian Wang** [5], **Peiying Zhang** [1,2], **Wei Zhang** [6] **and Leiquan Wang** [1]

1  College of Computer Science and Technology, China University of Petroleum (East China), Qingdao 266580, China; zhangjunsan@upc.edu.cn (J.Z.); z20070078@s.upc.edu.cn (L.Z.); zhangpeiying@upc.edu.cn (P.Z.); wangleiquan@upc.edu.cn (L.W.)
2  State Key Laboratory of Integrated Services Networks, Xidian University, Xi'an 710071, China
3  The Sixth Research Institute of China Electronics Corporation, Beijing 100083, China; lzhmjhz@163.com
4  School of Information Engineering, Huzhou University, Huzhou 313000, China
5  College of Science, China University of Petroleum (East China), Qingdao 266580, China; wangjiannl@upc.edu.cn
6  Shandong Provincial Key Laboratory of Computer Networks, Shandong Computer Science Center (National Supercomputer Center in Jinan), Qilu University of Technology (Shandong Academy of Sciences), Jinan 250013, China; wzhang@sdas.org
*  Correspondence: shigens@zjhu.edu.cn

**Abstract:** In recent years, hyperspectral image classification techniques have attracted a lot of attention from many scholars because they can be used to model the development of different cities and provide a reference for urban planning and construction. However, due to the difficulty in obtaining hyperspectral images, only a limited number of pixels can be used as training samples. Therefore, how to adequately extract and utilize the spatial and spectral information of hyperspectral images with limited training samples has become a difficult problem. To address this issue, we propose a hyperspectral image classification method based on dense pyramidal convolution and multi-feature fusion (DPCMF). In this approach, two branches are designed to extract spatial and spectral features, respectively. In the spatial branch, dense pyramid convolutions and non-local blocks are used to extract multi-scale local and global spatial features in image samples, which are then fused to obtain spatial features. In the spectral branch, dense pyramidal convolution layers are used to extract spectral features in image samples. Finally, the spatial and spectral features are fused and fed into fully connected layers to obtain classification results. The experimental results show that the overall accuracy (OA) of the method proposed in this paper is 96.74%, 98.10%, 98.92% and 96.67% on the four hyperspectral datasets, respectively. Significant improvements are achieved compared to the five methods of SVM, SSRN, FDSSC, DBMA and DBDA for hyperspectral classification. Therefore, the proposed method can better extract and exploit the spatial and spectral information in image samples when the number of training samples is limited. Provide more realistic and intuitive terrain and environmental conditions for urban planning, design, construction and management.

**Keywords:** hyperspectral image classification; image processing; spectral-spatial feature fusion; deep learning

## 1. Introduction

Hyperspectral images, also known as hyperspectral remote sensing images, are stereoscopic images captured by aerospace vehicles equipped with hyperspectral imagers. They consist of two spatial dimensions and one spectral dimension. The spectral dimension contains 10 s or even 100 s of spectral bands, which provide it with broad prospects for applications such as military target detection [1]; atmospheric and environmental research [2]; forest vegetation cover detection [3]; and change area detection [4]. Hyperspectral image

classification is a commonly used technique in the applications listed above. However, the excessive redundancy of spectral information and the limited number of training samples pose a great challenge for hyperspectral image classification.

In the early research on hyperspectral image classification, methods such as support vector machine (SVM) [5], multinomial logistic regression (MLR) [6] and sparse representation classification (SRC) [7] were proposed, which directly take the original input as the training sample and use it to train the classifier through the spectral information of the hyperspectral image. However, such methods ignore two problems: (1) the large amount of redundant information in spectral bands makes it difficult to train the model; (2) hyperspectral images have high spatial correlation and contain abundant spectral information. To solve problem (1), dimensionality reduction strategies [8,9] (feature selection [10] and feature extraction [11]) are applied to hyperspectral image classification tasks. To solve problem (2), morphological contours [12] and Gabor features [13] are used to ex-tract spatial information, and the morphological kernel [14] and composite kernel [15] methods are used to extract spectral–spatial information. Although the aforementioned methods improve the accuracy of the classifier, it is difficult to achieve better classification results in complex scenes because these methods use shallow models and rely heavily on labeled samples, which cannot extract the deep features of the samples.

Deep learning (DL) has shown strong capabilities in automatically extracting nonlinear and hierarchical features, and thus has been widely used in information extraction [16], image classification [17], semantic segmentation [18] and object detection [19]. Therefore, some hyperspectral image classification methods based on deep learning are proposed. In [20], Zhou et al. used a stacked auto-encoder (SAE) to extract spectral and spatial features and used logistic regression to obtain classification results. In [21], Szegedy C, et al. used a restricted Boltzmann machine (RBM) and deep belief network (DBN) for classification. In [22], Ma et al. used a spatially updated deep auto-encoder (DAE) to extract spectral–spatial features and designed a different co-representation mechanism to handle narrow-scale training sets. In [23], Zhang et al. utilized a recursive auto-encoder to learn the spatial and spectral information and adopted a weighting scheme to fuse the spatial information. Although these methods can extract the spectral–spatial features of hyperspectral images to a certain extent, they destroy the spatial structure. Since convolutional neural networks (CNNs) can exploit spatial features while preserving the original spatial structure, some methods based on CNNs have been proposed. In [24], Zhao et al. employed a CNN as the feature extractor. In [25], Zhang et al. proposed a method based on differentiated region convolutional neural network (DRCNN), which uses different image patches within the neighborhood of the target pixel as the input of the CNN, and the input data is effective reinforcement. In [26], Lee et al. proposed a contextual deep CNN (CDCNN) with deeper and wider network layers.

In general, deep-level features in the image can be extracted by increasing the depth of the network, but this also causes problems such as difficulty in model training and gradient vanishing. The residual network (ResNet) [27] and dense convolutional network (DenseNet) [28] solved this problem quite efficiently, and such networks can also extract deep-level features without increasing the depth of the network structure. Inspired by ResNet, the literature [29] proposed a spectral-spatial residual network (SSRN) that contains a spectral residual block and a spatial residual block for sequentially extracting spectral features and spatial features. Inspired by DenseNet, some literature [30] has proposed a fast dense spectral-spatial convolutional network (FDSSC), which achieves better performance while reducing the training time. Although the aforementioned methods solve the feature extraction problem using CNN, but in the process of model training, the attention of the convolutional layer to features is not the same. In order to optimize the extracted features, the attention mechanism is used to process different features differently. It is also a research hotspot in recent years. One study [31] proposed a feedback attention-based dense CNN network, while another [32] proposed a dual-branch multi-attention mechanism network (DBMA) based on the convolutional block attention module (CBAM) [33]. Moreover, [34]

proposed a dual-branch dual-attention mechanism network (DBDA) based on the dual-attention network (DANet) [35]. Although these methods are very effective, the extraction and utilization of spatial information and spectral information of hyperspectral images are not sufficient, resulting in the inability to obtain better classification results in the case of limited training samples.

Inspired by pyramidal convolution (PyConv) [36] and DenseNet, for the two problems of missing features and insufficient feature extraction, this paper proposes a hyperspectral image classification method based on dense pyramidal convolution and multi-feature fusion (DPCMF). The proposed method consists of two branches: the spatial branch and the spectral branch, both of which are designed to capture spectral and spatial features, respectively. In the spatial branch, principal component analysis (PCA) is performed to achieve dimensionality reduction for image samples, whereas noise and redundant spectral information are removed while retaining significant spectral information. Then, the dense pyramid convolution module and non-local block [37] are used to extract multi-scale local spatial information and global spatial information from image samples. Finally, the multi-scale local spatial information and global spatial information are fused to obtain a spatial feature map. In the spectral branch, convolutional neural networks are first used to perform convolutions on image samples, and then the spectral information in the images is extracted through dense pyramid convolution to obtain spectral feature maps. Lastly, the spatial and spectral feature maps are fused and fed into the classification module to obtain the classification results. The three main contributions of this paper are described below.

- To address the problem of missing features, a hyperspectral image classification method (DPCMF) based on dense pyramidal convolution and multi-feature fusion is proposed. This method uses spatial and spectral branches to extract local spatial features and spectral features, respectively. Multi-scale local spatial information is extracted using non-local block segmentation and fused to obtain hyperspectral feature maps.
- To address the problem of insufficient feature extraction, in the feature extraction part of the spatial and spectral branches, a combination of pyramidal convolution and DenseNet is used. Without increasing the depth of the network, the convolutional kernels are arranged in descending order to extract deep-level features at different scales.
- DPCMF achieves state-of-the-art classification accuracies on four datasets with limited training samples.

The rest of this paper is arranged as follows: Section 2 presents the specific implementation of the proposed method; Sections 3 and 4 present and analyze the experimental results; and Section 5 summarizes the conclusions of the article and proposes future directions for research.

## 2. Materials and Methods

In this section, we give a brief introduction to the structure of dense pyramidal convolution layers, non-local blocks and DPCMF networks.

### 2.1. Dense Pyramidal Convolution

In a traditional CNN, deeper features are generally extracted by increasing the network depth, but the problem of vanishing or exploding gradients occurs simultaneously, making it difficult to train deep models. DenseNet improves feature availability without increasing the network depth through dense connections, and it also solves the problem of gradient vanishing and gradient exploding. As shown in Figure 1, the purpose of feature reuse is achieved in DenseNet by building dense connections between all previous layers and subsequent layers to fuse shallow and deep features. The output $X_i$ of the $i$th layer can be expressed as:

$$X_i = H_i[X_0, X_1, \ldots, X_{i-1}] \tag{1}$$

where $H_i$ represents a nonlinear transformation including convolution, activation function, and batch normalization (BN), and [] means the output $X_0, X_1, .., X_{i-1}$ of the 0th layer to the $(i-1)$th layer are spliced along the channel dimension.

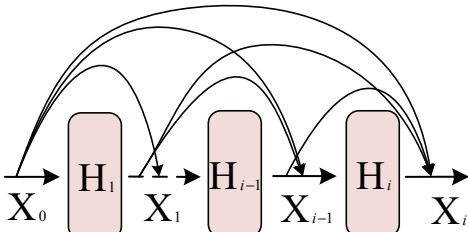

**Figure 1.** Architecture of DenseNet.

During the training process of DenseNet, convolution kernels of uniform size are normally used to extract image features, which will lead to insufficient actual receptive field of the convolutional neural network, inability to extract multi-scale information in image samples, and insufficient feature extraction. PyConv uses convolution kernels of different sizes to extract image features, which solves this problem effectively. Based on DenseNet and PyConv, we propose a dense pyramid structure as a feature extraction module for hyperspectral images. The three-dimensional convolutional neural network (3D-CNN) is obtained by adding a dimension of convolution calculation on basis of the original convolutional neural network, and the dimensionality of the convolution kernel is increased from the original two dimensions to three dimensions. Unlike 2D convolutions, which focus on exploring single feature information, 3D convolutions can slide through three dimensions, mining rich information from all feature maps. A hyperspectral image is a cubic structure that contains both spectral and spatial information, and 3D convolutions not only affect the spatial dimension, but they also involve multiple continuous spectral bands simultaneously. Therefore, spatial and spectral information can be obtained by applying 3D convolutional neural networks to extract features from hyperspectral images. As shown in Figure 2, 3D convolutions with convolution kernels arranged in descending order according to their sizes are used as the basic structure to extract images that are not used in the image samples. Scale information, using dense connections, concatenates shallow large-scale features and deep small-scale features to obtain a feature map that combines multi-scale information in image samples.

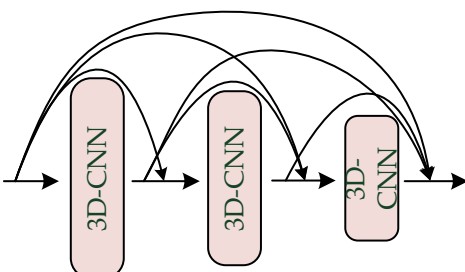

**Figure 2.** Architecture of the dense pyramidal convolution module.

The convolution process of the 3D convolutional layer is shown in Figure 3. The input is $n^k$ feature maps with a size of $p^k \times p^k \times b^k$, and after $n^{k+1}$ 3D-CNN layers with a certain convolution kernel size, $n^{k+1}$ feature maps with a size of $p^{k+1} \times p^{k+1} \times b^{k+1}$ are generated. The $i$th output of the $(k + 1)$th 3D-CNN layer is expressed as:

$$X_i^{k+1} = R\left(\sum_{j=1}^{n^k} \hat{X}_j^k \times H_i^{k+1} + c_i^{k+1}\right) \tag{2}$$

where $\hat{X}_j^k \in R^{p \times p \times k}$ represents the $j$th input feature map of the $(k+1)$th layer, $X_i^{k+1}$ represents the final output of the $k$th layer, $H_i^{k+1}$ and $c_i^{k+1}$ represent the weights and biases of the $(k+1)$th layer and R$(\cdot)$ represents the batch normalization (BN) and nonlinear activation function.

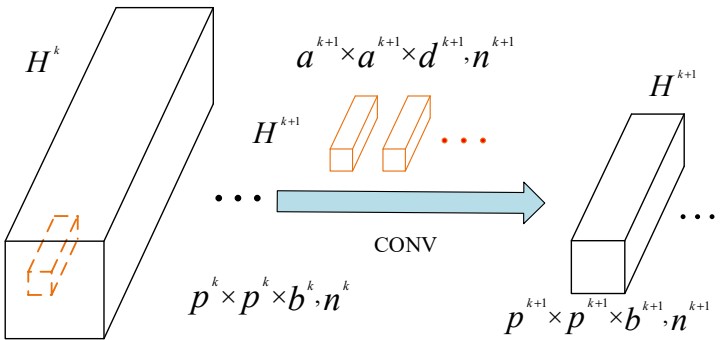

**Figure 3.** Architecture of a 3D-convolutional neural network (3D-CNN).

### 2.2. Non-Local Block

In a CNN, the convolution operation only focuses on the local receptive field. If the receptive field of a neuron is to be increased, this can only be achieved by stacking convolutional and pooling layers. However, this approach would considerably increase the computational load and the number of parameters. Therefore, in this paper, we use non-local blocks to capture the connections between distant pixels and obtain the global information in the image sample by computing the weights of all position features in the image sample. Figure 4 shows the structure of a non-local block, where X is the input feature map and Y is the output feature map with the same shape as X. θ, Φ, g and h represent convolution kernels with a size of $1 \times 1 \times 1$, where the number of θ, Φ and g is 1/2 of the number of channels of X and the number of h is the number of channels of X. The convolutional layer first reduces the number of channels of the input X to 1/2 of the original one through θ, Φ and g, thus reducing the computational load. This process can be expressed as:

$$\theta(x_i) = W_\theta x_i + b_\theta \tag{3}$$

$$\Phi(x_i) = W_\Phi x_i + b_\Phi \tag{4}$$

$$g(x_i) = W_g x_i + b_g \tag{5}$$

where $W$ and $b$ denote the weight matrix and bias in the convolution process, respectively. After three convolutions, the outputs $\theta(x_i)$, $\Phi(x_i)$ and $g(x_i)$ with the number of channels being 1/2 of that in $X$ are obtained, and the transposes of $\theta(x_i)$ and $\Phi(x_i)$ are point multiplication operations. This process can be expressed as:

$$f(x_i, x_j) = \theta(x_i)^T \Phi(x_j) \tag{6}$$

where $f(x_i, x_j)$ represents the influence of position $j$ on position $i$. The greater the value of $f$, the greater the influence of position $j$ on position $i$. The main idea is to perform point multiplication between the obtained output and the eigenvalue $g(x_j)$ of position j, restore the output through the convolution layer h until it has the same shape as the original input X and finally add the obtained weight matrix to the original input. This process can be expressed as:

$$y_i = f(x_i, x_j)g(x_j) \tag{7}$$

$$z_i = h(y_i) + x_i \tag{8}$$

where $y_i$ is the weight of position $i$ in the non-local block and $z_i$ denotes the final output of position $i$ in the non-local block. The final output Z of the entire non-local block can be expressed as:

$$Z = h\left(\frac{1}{C(x)}\sum_{\forall j} f(x_i, x_j) g(x_j)\right) + X \tag{9}$$

where $C(x)$ is a normalization parameter, and its value is the number of positions in *X*.

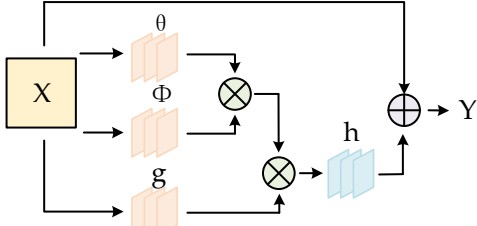

⊗ Matrix multiplication ⊕ Element-wise Sum

**Figure 4.** Architecture of the non-local block.

*2.3. Framework of the DPCMF Network*

The structure of the DPCMF network is shown in Figure 5. In this section, the Indian Pines (IP) dataset is taken as an example to describe the architecture of the DPCMF model. The Indian Pines dataset contains $145 \times 145$ pixels, and each pixel has 200 spectral bands. The size of this dataset is $145 \times 145 \times 200$, the number of pixels with labels is 20,249, the other pixels are the background, the number of label categories is 16, and the number of convolution kernels is 12.

In the spatial branch, the original image samples are processed to a size of $145 \times 145 \times 100$ by PCA, which reduces the amount of calculation and the number of model parameters while retaining the main information. Then, the image is split into training samples with a size of $9 \times 9 \times 100$. In the local information extraction module, each $9 \times 9 \times 100$ image sample is first convolved into a feature map with 24 channels and a size of $9 \times 9 \times 1$ through the convolution layer and then input into the spatial block that consists of three layers. The convolution kernels are arranged in descending order according to their sizes, which are $7 \times 7 \times 1$, $5 \times 5 \times 1$, and $3 \times 3 \times 1$. Each convolutional layer is followed by a batch normalization layer and a ReLU [38] activation function, and finally, a feature map with a channel number of 60 and a size of $9 \times 9 \times 1$ is obtained. In the global information extraction module, the feature map with the same shape as the input is obtained through the non-local module and then input into the convolutional layer to obtain a feature map with a channel number of 60 and a size of $9 \times 9 \times 1$. After the global and local information in the image samples is extracted by the two modules, the feature maps of the two parts are concatenated to obtain the spatial feature maps. The specific implementation details of the spatial branch are given in Table 1.

**Table 1.** Implementation details of the spatial branch.

| Layer Name | Kernel Size | Output Size |
|---|---|---|
| PCA | — | $(145 \times 145 \times 100)$ |
| Conv | $(3 \times 3 \times 1)$ | $(9 \times 9 \times 1, 24)$ |
| Spatial Block-1 | $(7 \times 7 \times 1)$ | $(9 \times 9 \times 1, 12)$ |
| Spatial Block-2 | $(5 \times 5 \times 1)$ | $(9 \times 9 \times 1, 12)$ |
| Spatial Block-3 | $(3 \times 3 \times 1)$ | $(9 \times 9 \times 1, 12)$ |
| Non-local block | — | $(9 \times 9 \times 100)$ |
| Conv | $(3 \times 3 \times 1)$ | $(9 \times 9 \times 1, 60)$ |
| Concatenate | — | $(9 \times 9 \times 1, 120)$ |

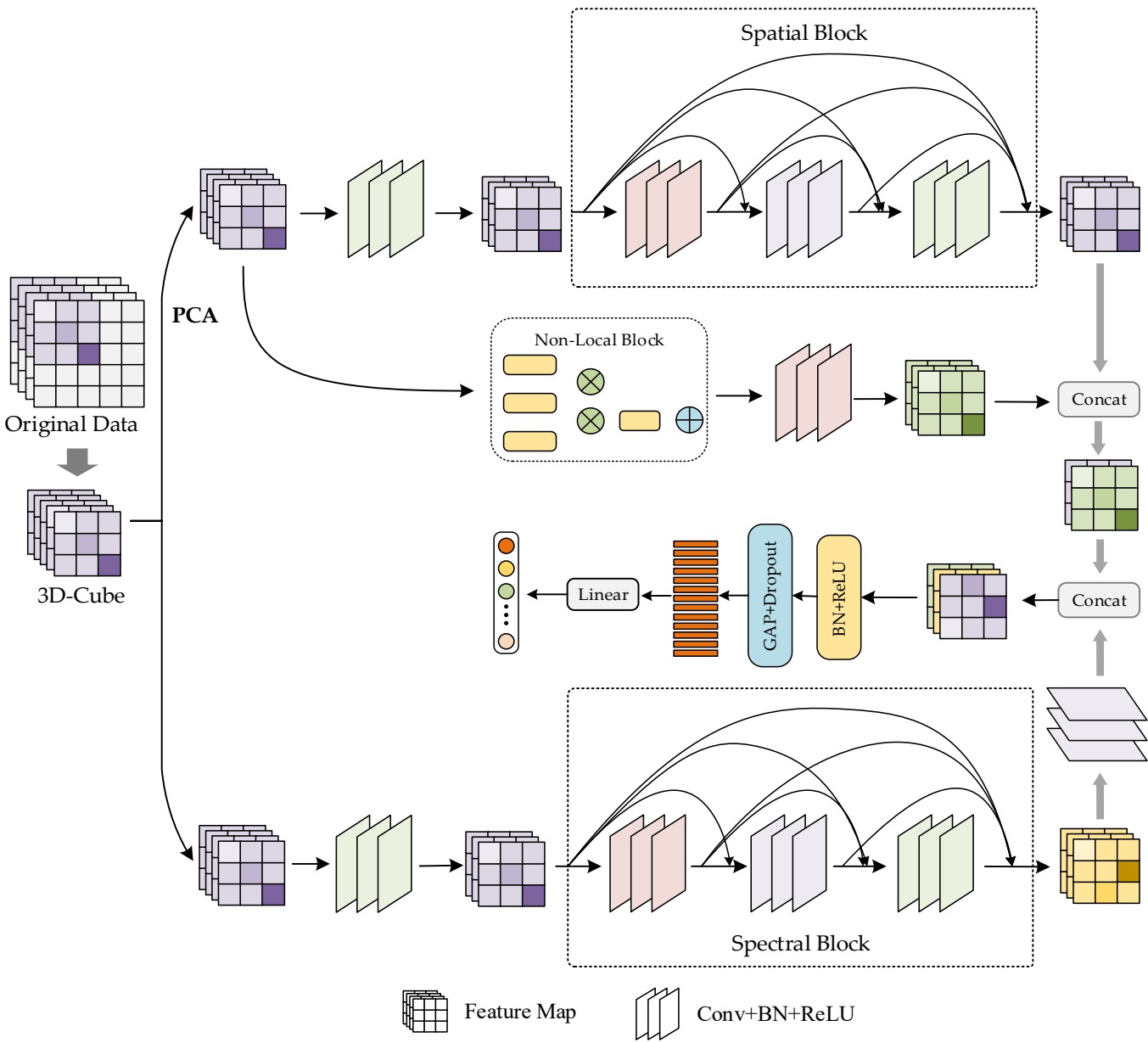

**Figure 5.** Structure of the DPCMF network.

In the spectral branch, the image is split into training samples with a size of $9 \times 9 \times 200$, each sample is first convolved into a $9 \times 9 \times 97$ feature map through the convolutional layer and then input into the spectral block. The spectral block consists of three convolutional layers. The convolutional kernels are arranged in descending order according to their sizes, which are $1 \times 1 \times 7$, $1 \times 1 \times 5$ and $1 \times 1 \times 3$, respectively. After each convolutional layer, a batch normalization layer and a ReLU activation function are added, and the image sample is input to the convolution kernel in the convolutional layer with a size of $1 \times 1 \times 97$ to obtain the spectral feature map. The specific implementation details of the spectral branch are listed in Table 2.

In the classification module, the spatial feature map obtained by the spatial branch and the spectral feature map obtained by the spectral branch are subjected to feature fusion to obtain a feature map with a channel number of 180 and a size of $9 \times 9 \times 1$; then, a feature map with a size of $1 \times 180$ is obtained through the global average pooling (GAP) layer; finally, the classification result is obtained through the fully connected layer and the

Softmax activation function. The implementation details of the classification module are given in Table 3.

**Table 2.** Implementation details of the spectral branch.

| Layer Name | Kernel Size | Output Size |
|---|---|---|
| Conv | $(1 \times 1 \times 7)$ | $(9 \times 9 \times 97)$ |
| Spectral Block-1 | $(1 \times 1 \times 7)$ | $(9 \times 9 \times 97, 12)$ |
| Spectral Block-2 | $(1 \times 1 \times 5)$ | $(9 \times 9 \times 97, 12)$ |
| Spectral Block-3 | $(1 \times 1 \times 3)$ | $(9 \times 9 \times 97, 12)$ |
| Conv | $(1 \times 1 \times 97)$ | $(9 \times 9 \times 1, 60)$ |

**Table 3.** Implementation details of the classification module.

| Layer Name | Kernel Size | Output Size |
|---|---|---|
| Concatenation layer | — | $(9 \times 9 \times 1, 180)$ |
| GAP layer | — | $(1 \times 180)$ |
| Fully Connected layer | — | $(1 \times 16)$ |

## 3. Experimental Results

In order to verify the effectiveness of the method proposed in this paper, four public hyperspectral datasets, namely, Indian Pines (IP), Pavia University (UP), Salinas Valley (SV) and Botswana (BS), are used to conduct experiments. The accuracy of each method is measured by three evaluation indicators: overall accuracy (OA), average accuracy (AA) and Kappa coefficient. OA represents the proportion of correctly classified samples to the total test samples, and AA represents the average accuracy across all categories. The Kappa coefficient represents the level of consistency between the true value and the classification result. The greater the values of these three evaluation metrics, the better the classification results.

### 3.1. Introduction and Division of the Dataset

Indian Pines (IP): Indian Pines was imaged by the Airborne Visible Infrared Imaging Spectrometer (AVIRIS) in 1992 on an Indian pine tree in Indiana, USA and marked with a size of $145 \times 145$. The imaging wavelength range of AVIRIS is 0.4–2.5 μm, and it continuously images ground objects in 220 continuous bands. However, because the 104th–108th, 150th–163rd and 220th bands cannot be reflected by water, 200 bands covering 16 types of ground objects are truly used for training.

Pavia University (UP): Pavia University is part of the hyperspectral data of the city of Pavia in Italy in 2003 collected by the German Airborne Reflective Optics Spectrographic Imaging System (ROSIS-03). The size of this dataset is $610 \times 340$. The spectral imager continuously captures images with 115 bands within the wavelength range of 0.43–0.86 μm, and the spatial resolution of the resulting images is 1.3 m. Among these bands, 12 bands were eliminated due to noise effects, and 103 bands covering 9 types of ground objects were used for real-world training.

Salinas Valley (SV): The Salinas Valley dataset was acquired by the AVIRIS sensor in Salinas Valley, California. The size of this dataset is $512 \times 217$, the spatial resolution is 3.7 m, and it contains 224 continuous bands. In total, 20 water-absorbing bands (108–112, 154–167, 224) were removed, and 204 bands covering 16 types of ground objects were actually used for training.

Botswana (BS): The Botswana dataset was acquired by the NASA EO-1 satellite in the Okavango Delta of Botswana in May 2001. The size of this dataset is $1476 \times 256$. The sensor on EO-1 has a wavelength range of 400–2500 nm and a spatial resolution of about 20 m. Among the 242 bands, the noise bands (1–9, 56–81, 98–101, 120–133, 165–186) were removed, and 145 bands covering 14 types of ground objects were actually used for training.

Before conducting the experiments, we split each dataset into three parts, namely, training set, validation set, and test set. The training set is used to update model parameters, the validation set is used to monitor the temporary models generated during the training phase, and the test set is used to evaluate the optimal model. For different datasets, the proportions of the three parts are different. The division of Indian Pines (IP) is shown in Table 4, the division of Pavia University (UP) is shown in Table 5, the division of Salinas Valley (SV) is shown in Table 6 and the division of Botswana (BS) is shown in Table 7.

**Table 4.** Samples of the IP dataset for training, validation and testing.

| Order | Class | Total | Train | Val. | Test |
|---|---|---|---|---|---|
| 1 | Alfalfa | 46 | 3 | 3 | 40 |
| 2 | Corn-notill | 1428 | 42 | 42 | 1344 |
| 3 | Corn-mintill | 830 | 24 | 24 | 782 |
| 4 | Corn | 237 | 7 | 7 | 223 |
| 5 | Grass-pasture | 483 | 14 | 14 | 455 |
| 6 | Grass-trees | 730 | 21 | 21 | 688 |
| 7 | Grass-pasture-mowed | 28 | 3 | 3 | 22 |
| 8 | Hay-windrowed | 478 | 14 | 14 | 450 |
| 9 | Oats | 20 | 3 | 3 | 14 |
| 10 | Soybean-notill | 972 | 29 | 29 | 914 |
| 11 | Soybean-mintill | 2455 | 73 | 73 | 2309 |
| 12 | Soybean-clean | 593 | 17 | 17 | 559 |
| 13 | Wheat | 205 | 6 | 6 | 193 |
| 14 | Woods | 1265 | 37 | 37 | 1191 |
| 15 | Buildings-Grass-Tree-Drives | 386 | 11 | 11 | 364 |
| 16 | Stone-Steel-Towers | 93 | 3 | 3 | 87 |
| | Total | 10249 | 307 | 307 | 9635 |

**Table 5.** Samples of the UP dataset for training, validation and testing.

| Order | Class | Total | Train | Val. | Test |
|---|---|---|---|---|---|
| 1 | Asphalt | 6631 | 33 | 33 | 6465 |
| 2 | Meadows | 18,649 | 93 | 93 | 18,463 |
| 3 | Gravel | 2099 | 10 | 10 | 2079 |
| 4 | Corn | 3064 | 15 | 15 | 3034 |
| 5 | Trees | 1345 | 6 | 6 | 1333 |
| 6 | Bare Soil | 5029 | 25 | 25 | 4979 |
| 7 | Bitumen | 1330 | 6 | 6 | 1318 |
| 8 | Self-Blocking Bricks | 3682 | 18 | 18 | 3646 |
| 9 | Shadows | 947 | 4 | 4 | 939 |
| | Total | 42,776 | 210 | 210 | 42,356 |

**Table 6.** Samples of the SV dataset for training, validation and testing.

| Order | Class | Total | Train | Val. | Test |
|---|---|---|---|---|---|
| 1 | Brocoli-green-weeds-1 | 2009 | 10 | 10 | 1989 |
| 2 | Brocoli-green-weeds-2 | 3726 | 18 | 18 | 3690 |
| 3 | Fallow | 1976 | 9 | 9 | 1958 |
| 4 | Fallow-rough-plow | 1394 | 6 | 6 | 1382 |
| 5 | Fallow-smooth | 2678 | 13 | 13 | 2652 |
| 6 | Stubble | 3959 | 19 | 19 | 3921 |
| 7 | Celery | 3579 | 17 | 17 | 3545 |
| 8 | Grapes-untrained | 11,271 | 56 | 56 | 11,159 |
| 9 | Soil-vineyard-develop | 6203 | 31 | 31 | 6141 |
| 10 | Corn-senesced-green-weeds | 3278 | 16 | 16 | 3246 |
| 11 | Lettuce-romaine-4wk | 1068 | 5 | 5 | 1058 |
| 12 | Lettuce-romaine-5wk | 1927 | 9 | 9 | 1909 |

**Table 6.** *Cont.*

| Order | Class | Total | Train | Val. | Test |
|-------|-------|-------|-------|------|------|
| 13 | Lettuce-romaine-6wk | 916 | 4 | 4 | 908 |
| 14 | Lettuce-romaine-7wk | 1070 | 5 | 5 | 1060 |
| 15 | Vineyard-untrained | 7268 | 36 | 36 | 7196 |
| 16 | Vineyard-vertical-trellis | 1807 | 9 | 9 | 1789 |
| | Total | 54,129 | 263 | 263 | 53,603 |

**Table 7.** Samples of the BS dataset for training, validation and testing.

| Order | Class | Total | Train | Val. | Test |
|-------|-------|-------|-------|------|------|
| 1 | Water | 270 | 3 | 3 | 264 |
| 2 | Hippo grass | 101 | 2 | 2 | 97 |
| 3 | Floodplain grasses1 | 251 | 3 | 3 | 245 |
| 4 | Floodplain grasses2 | 215 | 3 | 3 | 209 |
| 5 | Reeds1 | 269 | 3 | 3 | 263 |
| 6 | Riparian | 269 | 3 | 3 | 263 |
| 7 | Fierscar2 | 259 | 3 | 3 | 253 |
| 8 | Island interior | 203 | 3 | 3 | 197 |
| 9 | Acacia woodlands | 314 | 4 | 4 | 306 |
| 10 | Acacia shrublands | 248 | 3 | 3 | 242 |
| 11 | Acacia grasslands | 305 | 4 | 4 | 297 |
| 12 | Short mopane | 181 | 2 | 2 | 177 |
| 13 | Mixed mopane | 269 | 3 | 3 | 263 |
| 14 | Exposed soils | 95 | 1 | 1 | 93 |
| | Total | 3248 | 40 | 40 | 3168 |

*3.2. Experimental Setting*

To validate the classification performance of the, we conducted experiments to compare the DPCMF network with the SVM, SSRN, FDSSC, DBMA and DBDA classification networks. All experiments were performed on Intel (R) Xeon (R) 4208 CPU @ 2.10 GHz processor with Nvidia GeForce RTX Running on the 2060Ti graphics card system. The programming language used is Python. All classification networks were implemented using PyTorch, PyCharm was used as the compiler, the batch size was set to 16, RMSprop was used as the optimizer, the initial learning rate was set to 0.00005, and the cross-entropy loss function was used for experiments.

*3.3. Classification Maps and Categorized Results*

3.3.1. Classification Maps and Categorized Results for the IP Dataset

In this experiment, 3% of the samples were used as training samples, 3% as validation samples, and 94% as test samples. The categorized results of different methods on the IP dataset are listed in Table 8, and the classification maps are shown in Figure 6.

**Table 8.** Categorized results for the IP dataset with 3% training samples.

| Class | SVM | SSRN | FDSSC | DBMA | DBDA | DPCMF |
|-------|-----|------|-------|------|------|-------|
| 1/% | 24.19 | 60.00 | 97.67 | 83.33 | 97.72 | 100 |
| 2/% | 56.71 | 91.47 | 99.12 | 92.27 | 96.43 | 97.62 |
| 3/% | 65.09 | 93.51 | 95.85 | 92.37 | 97.81 | 95.83 |
| 4/% | 39.63 | 88.95 | 100 | 100 | 97.56 | 100 |
| 5/% | 87.33 | 100 | 98.35 | 98.24 | 98.30 | 92.86 |
| 6/% | 83.87 | 95.95 | 88.44 | 98.40 | 96.75 | 95.24 |
| 7/% | 57.50 | 86.20 | 82.75 | 39.59 | 88.00 | 90.26 |
| 8/% | 89.28 | 94.50 | 100 | 99.10 | 100 | 100 |
| 9/% | 22.58 | 69.56 | 93.33 | 26.31 | 100 | 100 |

**Table 8.** *Cont.*

| Class | SVM | SSRN | FDSSC | DBMA | DBDA | DPCMF |
|---|---|---|---|---|---|---|
| 10/% | 66.70 | 84.35 | 87.31 | 83.98 | 91.12 | 93.10 |
| 11/% | 62.50 | 91.86 | 99.09 | 95.65 | 98.63 | 97.26 |
| 12/% | 51.86 | 86.74 | 89.01 | 85.05 | 93.55 | 94.12 |
| 13/% | 94.79 | 98.97 | 98.92 | 100 | 97.97 | 100 |
| 14/% | 90.42 | 94.74 | 96.24 | 93.73 | 94.40 | 97.30 |
| 15/% | 62.82 | 95.09 | 94.41 | 94.37 | 92.98 | 90.91 |
| 16/% | 98.46 | 91.11 | 94.38 | 96.51 | 95.45 | 93.44 |
| OA/% | 69.35 | 91.95 | 95.45 | 92.85 | 96.13 | 96.74 |
| AA/% | 65.86 | 88.94 | 94.68 | 86.18 | 95.69 | 96.12 |
| Kappa/% | 64.65 | 90.81 | 94.82 | 91.85 | 95.43 | 96.32 |
| Training Time/s | 11.85 | 52.30 | 128.74 | 113.25 | 77.25 | 72.41 |
| Test Time/s | 1.62 | 3.12 | 6.32 | 8.96 | 7.02 | 7.25 |

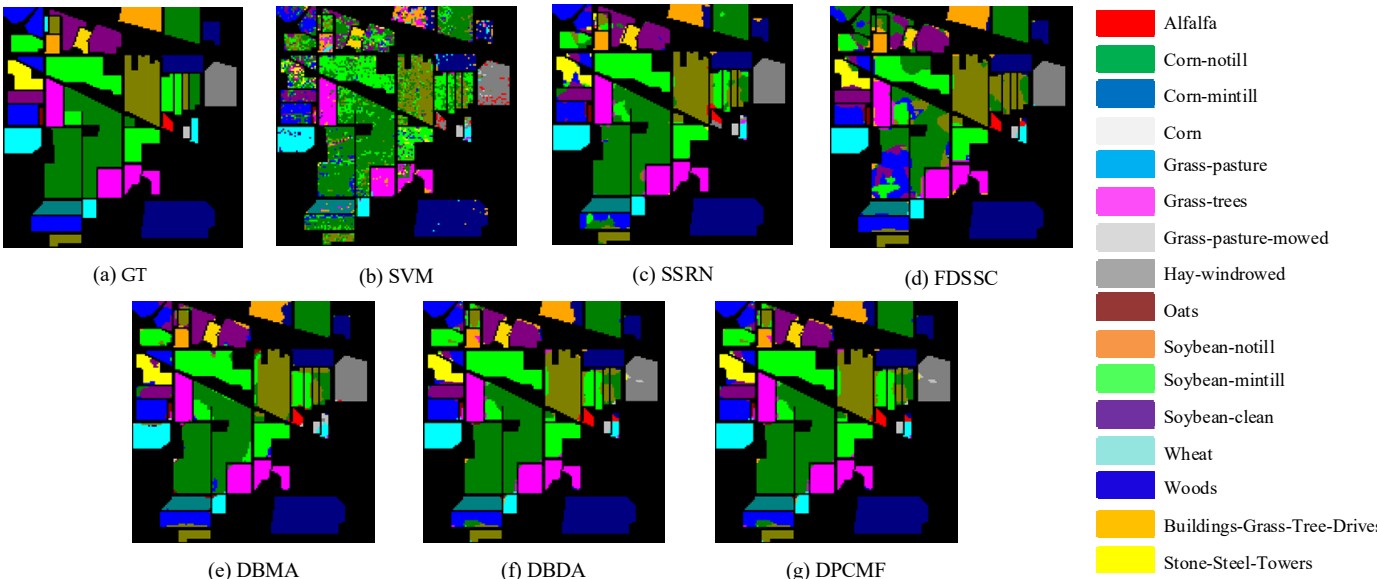

**Figure 6.** Classification maps for the IP dataset with 3% training samples. (**a**) Ground-truth (GT); (**b**–**g**) classification maps from disparate algorithms.

### 3.3.2. Classification Maps and Categorized Results for the UP Dataset

In this experiment, 0.5% of the samples were used as training samples, 0.5% as validation samples, and 99% as test samples. The categorized results of different methods on the UP dataset are listed in Table 9, and the classification maps are shown in Figure 7.

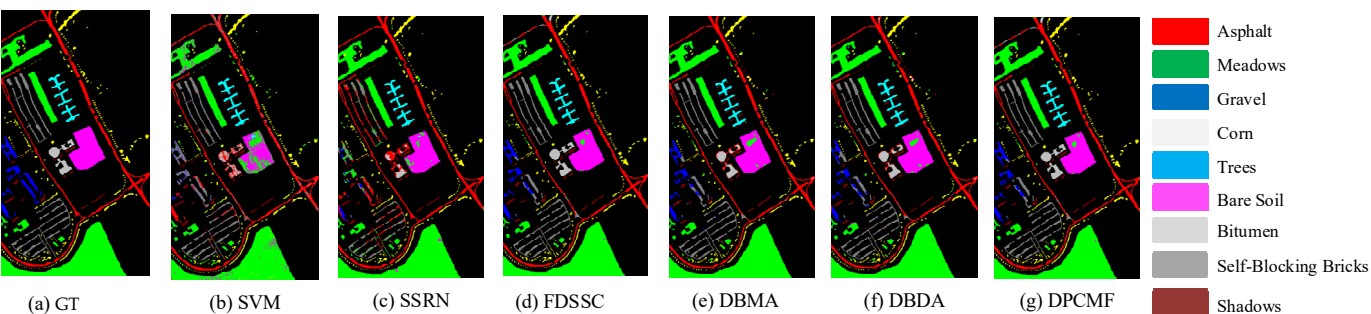

**Figure 7.** Classification maps for the UP dataset with 0.5% training samples. (**a**) Ground-truth (GT); (**b**–**g**) classification maps from disparate algorithms.

**Table 9.** Categorized results for the UP dataset with 0.5% training samples.

| Class | SVM | SSRN | FDSSC | DBMA | DBDA | DPCMF |
|-------|-----|------|-------|------|------|-------|
| 1/% | 80.26 | 68.45 | 99.55 | 93.49 | 95.27 | 96.97 |
| 2/% | 86.94 | 92.95 | 98.57 | 96.98 | 96.81 | 98.92 |
| 3/% | 71.73 | 99.52 | 100 | 96.72 | 99.26 | 100 |
| 4/% | 96.44 | 99.08 | 98.07 | 96.93 | 98.20 | 96.67 |
| 5/% | 90.85 | 97.77 | 99.70 | 99.62 | 98.61 | 100 |
| 6/% | 77.02 | 98.15 | 99.81 | 99.52 | 98.83 | 98.00 |
| 7/% | 69.70 | 100 | 94.74 | 97.11 | 100 | 100 |
| 8/% | 67.30 | 83.12 | 80.70 | 82.25 | 89.09 | 94.44 |
| 9/% | 99.89 | 99.25 | 99.88 | 98.59 | 100 | 100 |
| OA/% | 83.07 | 88.32 | 96.92 | 95.28 | 97.12 | 98.10 |
| AA/% | 82.24 | 93.14 | 96.78 | 95.69 | 97.34 | 98.33 |
| Kappa/% | 77.07 | 84.14 | 95.91 | 93.70 | 96.23 | 97.77 |
| Training Time/s | 5.69 | 11.88 | 31.38 | 10.65 | 21.21 | 21.08 |
| Test Time/s | 2.09 | 5.21 | 13.56 | 13.12 | 11.05 | 12.98 |

### 3.3.3. Classification Maps and Categorized Results for the SV Dataset

In this experiment, 0.5% of the samples were used as training samples, 0.5% as validation samples, and 99% as test samples. The categorized results of different methods on the SV dataset are listed in Table 10, and the classification maps are shown in Figure 8.

**Table 10.** Categorized results for the SV dataset with 0.5% training samples.

| Class | SVM | SSRN | FDSSC | DBMA | DBDA | DPCMF |
|-------|-----|------|-------|------|------|-------|
| 1/% | 99.84 | 99.34 | 100 | 100 | 100 | 100 |
| 2/% | 98.95 | 100 | 97.20 | 100 | 99.62 | 100 |
| 3/% | 89.87 | 87.71 | 99.28 | 98.38 | 100 | 100 |
| 4/% | 97.30 | 95.22 | 96.23 | 92.02 | 89.35 | 91.67 |
| 5/% | 93.55 | 99.54 | 99.96 | 99.11 | 99.63 | 100 |
| 6/% | 99.79 | 99.79 | 99.92 | 99.84 | 99.26 | 100 |
| 7/% | 91.33 | 99.52 | 100 | 98.74 | 96.69 | 100 |
| 8/% | 74.73 | 84.51 | 99.80 | 93.15 | 97.95 | 98.21 |
| 9/% | 97.69 | 99.67 | 99.09 | 99.23 | 99.07 | 100 |
| 10/% | 90.01 | 99.28 | 99.17 | 97.42 | 93.76 | 100 |
| 11/% | 75.92 | 94.04 | 94.37 | 80.71 | 92.89 | 96.00 |
| 12/% | 95.19 | 96.96 | 99.84 | 99.44 | 100 | 100 |
| 13/% | 94.86 | 100 | 100 | 99.44 | 100 | 100 |
| 14/% | 89.26 | 98.22 | 99.06 | 96.33 | 95.81 | 97.00 |
| 15/% | 75.85 | 92.10 | 83.81 | 97.18 | 93.48 | 97.22 |
| 16/% | 99.03 | 100 | 100 | 100 | 100 | 100 |
| OA/% | 88.09 | 94.42 | 96.52 | 96.95 | 97.31 | 98.92 |
| AA/% | 91.45 | 96.62 | 97.81 | 96.97 | 97.34 | 98.76 |
| Kappa/% | 86.70 | 93.77 | 95.38 | 96.60 | 97.81 | 98.73 |
| Training Time/s | 10.27 | 84.99 | 123.28 | 149.12 | 81.20 | 78.22 |
| Test Time/s | 4.13 | 16.32 | 31.05 | 41.33 | 23.60 | 23.66 |

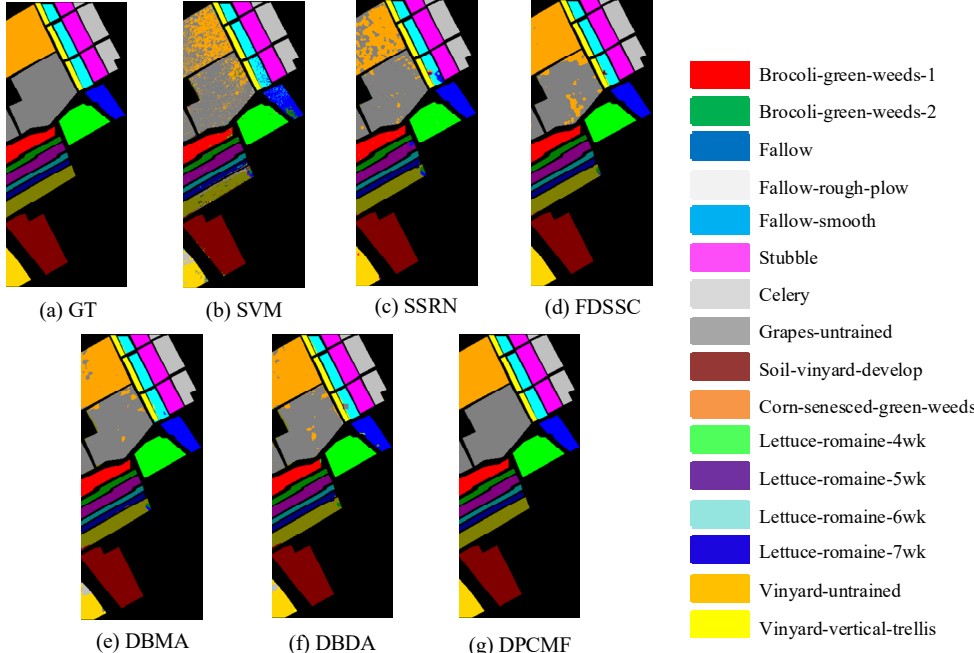

**Figure 8.** Classification maps for the SV dataset with 0.5% training samples. (**a**) Ground-truth (GT); (**b**–**g**) classification maps from disparate algorithms.

### 3.3.4. Classification Maps and Categorized Results for the BS Dataset

In this experiment, 1.2% of the samples were used as training samples, 1.2% as validation samples, and 97.6% as test samples. The categorized results of the different methods on the BS dataset are listed in Table 11, and the classification maps are shown in Figure 9.

**Table 11.** Categorized results for the BS dataset with 1.2% training samples.

| Class | SVM | SSRN | FDSSC | DBMA | DBDA | DPCMF |
|---|---|---|---|---|---|---|
| 1/% | 100 | 98.47 | 94.02 | 97.76 | 95.97 | 97.76 |
| 2/% | 70.70 | 94.62 | 100 | 98.98 | 98.00 | 100 |
| 3/% | 84.10 | 87.89 | 100 | 100 | 100 | 100 |
| 4/% | 65.95 | 86.80 | 96.89 | 89.40 | 85.77 | 88.28 |
| 5/% | 82.62 | 74.50 | 87.50 | 92.27 | 93.96 | 94.36 |
| 6/% | 65.71 | 80.19 | 69.76 | 80.13 | 87.04 | 88.65 |
| 7/% | 78.77 | 90.35 | 100 | 96.93 | 100 | 99.21 |
| 8/% | 65.87 | 87.11 | 95.60 | 100 | 99.32 | 100 |
| 9/% | 75.18 | 93.76 | 100 | 94.42 | 91.04 | 100 |
| 10/% | 69.82 | 81.56 | 91.04 | 92.77 | 100 | 89.7 |
| 11/% | 95.49 | 100 | 100 | 100 | 100 | 100 |
| 12/% | 93.10 | 100 | 88.88 | 100 | 100 | 100 |
| 13/% | 76.25 | 96.25 | 100 | 100 | 100 | 100 |
| 14/% | 90.41 | 100 | 100 | 97.43 | 100 | 100 |
| OA/% | 78.63 | 90.26 | 93.17 | 95.19 | 96.39 | 96.67 |
| AA/% | 79.57 | 90.82 | 94.45 | 95.72 | 96.50 | 97.08 |
| Kappa/% | 76.87 | 89.47 | 92.59 | 94.79 | 96.09 | 96.57 |
| Training Time/s | 1.62 | 11.23 | 23.02 | 21.09 | 17.96 | 20.39 |
| Test Time/s | 0.38 | 2.01 | 2.65 | 3.11 | 2.10 | 2.32 |

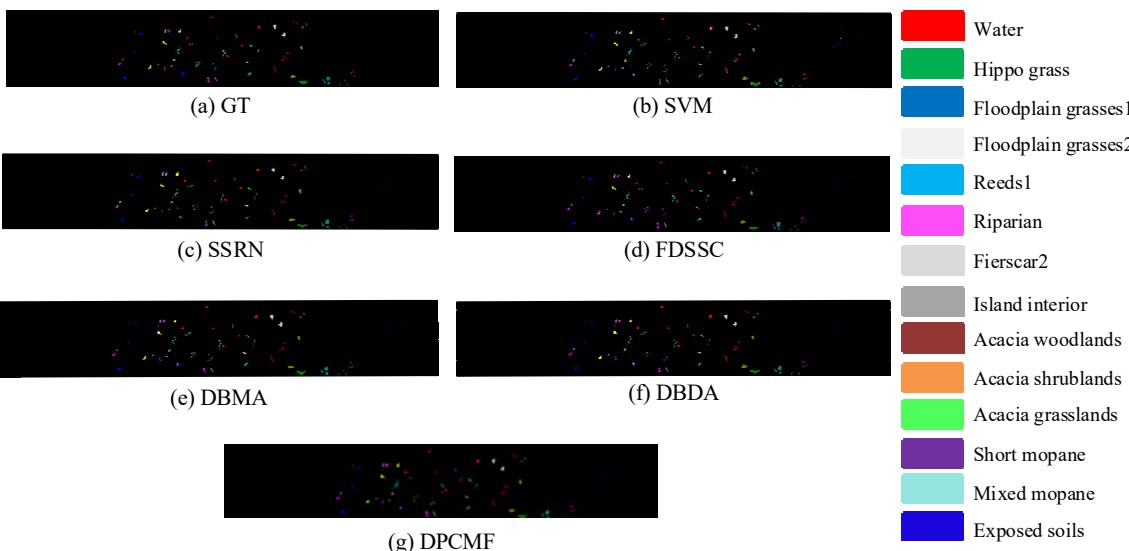

**Figure 9.** Classification maps for the BS dataset with 1.2% training samples. (**a**) Ground-truth (GT); (**b–g**) classification maps from disparate algorithms.

### 3.4. Impact of Convolution Kernel Size

In the process of feature extraction, the size of the convolution kernel affects the degree to which information is extracted. In this chapter experiment, convolution kernels of different sizes were used to extract features from hyperspectral images. To evaluate the impact of the convolution kernel size on the experimental results, experiments were conducted using convolution kernels of the same size instead of Pyconv. The experimental results are shown in Table 12. In the table, DPCMF_3 represents the use of a $3 \times 3 \times 1$ convolution kernel in the spatial block and a $1 \times 1 \times 3$ convolution kernel in the spectral block; DPCMF_5 represents the use of a $5 \times 5 \times 1$ convolution kernel in the spatial block and a $1 \times 1 \times 5$ convolution kernel in the spectral block; DPCMF_7 represents the use of a $7 \times 7 \times 1$ convolution kernel in the spatial block and a $1 \times 1 \times 7$ convolution kernel in the spectral block; and DPCMF_9 represents the use of a $9 \times 9 \times 1$ convolution kernel in the spatial block and a $1 \times 1 \times 9$ convolution kernel in the spectral block.

**Table 12.** Impact of kernel size on OA.

| Kernel Size | IP | UP | SV | BS |
|---|---|---|---|---|
| DPCMF_3/% | 89.13 | 93.15 | 92.12 | 90.61 |
| DPCMF-5/% | 93.28 | 96.22 | 95.42 | 94.22 |
| DPCMF-7/% | 95.12 | 97.02 | 97.38 | 94.67 |
| DPCMF-9/% | 94.37 | 96.58 | 96.88 | 95.13 |

### 3.5. Impact of Training Sample Size

To verify the impact of the number of training samples on the experimental results, experiments were conducted using a varying number of samples as training samples. For the IP dataset, 0.5%, 1%, 3%, 5% and 10% of the samples were used as training sets, and the experimental results are shown in Table 13. For the UP and SV datasets, 0.1%, 0.5%, 1%, 3% and 5% of the samples were used as training sets, and the experimental results are shown in Tables 14 and 15, respectively. For the BS dataset, 0.5%, 1.2%, 3%, 5% and 10% of the samples were used as training sets, and the experimental results are shown in Table 16.

**Table 13.** OA for different proportions of training samples in the IP dataset.

| Algorithms | 0.5% | 1% | 3% | 5% | 10% |
|---|---|---|---|---|---|
| SVM/% | 48.53 | 55.95 | 69.35 | 74.74 | 80.55 |
| SSRN/% | 64.99 | 81.40 | 90.52 | 0.955 | 97.84 |
| FDSSC/% | 70.75 | 84.71 | 96.14 | 97.21 | 98.02 |
| DBMA/% | 59.33 | 77.64 | 93.14 | 93.75 | 96.91 |
| DBDA/% | 56.97 | 78.81 | 96.19 | 96.58 | 97.55 |
| DPCMF/% | 73.25 | 85.14 | 96.74 | 97.95 | 98.56 |

**Table 14.** OA for different proportions of training samples in the UP dataset.

| Algorithms | 0.1% | 0.5% | 1% | 3% | 5% |
|---|---|---|---|---|---|
| SVM/% | 70.59 | 83.07 | 88.45 | 90.35 | 93.29 |
| SSRN/% | 78.32 | 94.85 | 97.11 | 99.43 | 99.69 |
| FDSSC/% | 88.97 | 97.02 | 97.74 | 99.50 | 99.58 |
| DBMA/% | 89.87 | 95.06 | 96.37 | 99.10 | 99.49 |
| DBDA/% | 88.01 | 97.11 | 98.40 | 99.07 | 99.33 |
| DPCMF/% | 91.35 | 98.10 | 98.89 | 99.99 | 99.99 |

**Table 15.** OA for different proportions of training samples in the SV dataset.

| Algorithms | 0.1% | 0.5% | 1% | 3% | 5% |
|---|---|---|---|---|---|
| SVM/% | 78.65 | 88.09 | 89.89 | 91.24 | 92.47 |
| SSRN/% | 67.22 | 95.35 | 96.32 | 97.23 | 98.14 |
| FDSSC/% | 88.83 | 95.85 | 96.48 | 97.52 | 98.85 |
| DBMA/% | 92.15 | 95.90 | 96.66 | 97.62 | 98.21 |
| DBDA/% | 94.23 | 97.70 | 98.31 | 98.95 | 99.36 |
| DPCMF/% | 96.04 | 98.92 | 99.25 | 99.90 | 99.99 |

**Table 16.** OA for different proportions of training samples in the BS dataset.

| Algorithms | 0.5% | 1.2% | 3% | 5% | 10% |
|---|---|---|---|---|---|
| SVM/% | 73.53 | 78.63 | 87.82 | 89.06 | 92.76 |
| SSRN/% | 84.07 | 94.27 | 95.52 | 98.19 | 99.15 |
| FDSSC/% | 87.98 | 90.80 | 96.33 | 97.24 | 99.46 |
| DBMA/% | 93.36 | 94.87 | 95.88 | 98.01 | 99.04 |
| DBDA/% | 96.27 | 96.39 | 97.38 | 98.64 | 99.33 |
| DPCMF/% | 96.37 | 96.67 | 99.10 | 99.58 | 99.91 |

*3.6. Ablation Experiment*

To verify the impact of the spatial block, spectral block, and non-local block on OA, experiments were conducted on these three modules using four datasets, as shown in Table 17. Table 17 displays the classification results of DPCMF, DPCMF-AE, DPCMF-AN, DPCM-EN and DPCMF-D on the four datasets. DPCMF-AE represents the absence of the non-local block, DPCMF-AN represents the absence of the spectral block, DPCMF-EN represents the absence of the spatial block, and DPCMF-D represents the absence of the Dense structure. The dataset partitioning process is consistent with that described in the previous section.

**Table 17.** Ablation experiments on four datasets.

| Algorithms | IP | UP | SV | BS |
|:---:|:---:|:---:|:---:|:---:|
| DPCMF/% | 96.74 | 98.10 | 98.92 | 96.67 |
| DPCMF-AE/% | 94.25 | 95.16 | 95.42 | 92.19 |
| DPCMF-AN/% | 90.38 | 90.30 | 91.37 | 89.24 |
| DPCMF-EN/% | 82.13 | 82.97 | 85.21 | 81.98 |
| DPCMF-D/% | 95.11 | 94.26 | 93.28 | 90.56 |

## 4. Discussion

The experimental results are shown in Figure 10a, DPCMF method achieves significant improvements in all of the following three metrics: OA, AA and Kappa. In terms of time, due to the large input volume of convolutional neural networks and the need for more training parameters, the time cost of the DPCMF method is higher than that of the SVM method, but in terms of classification accuracy, the accuracy level of the SVM method is much lower than those of other deep learning methods. In most cases, the DPCMF method takes less time than other deep learning-based methods.

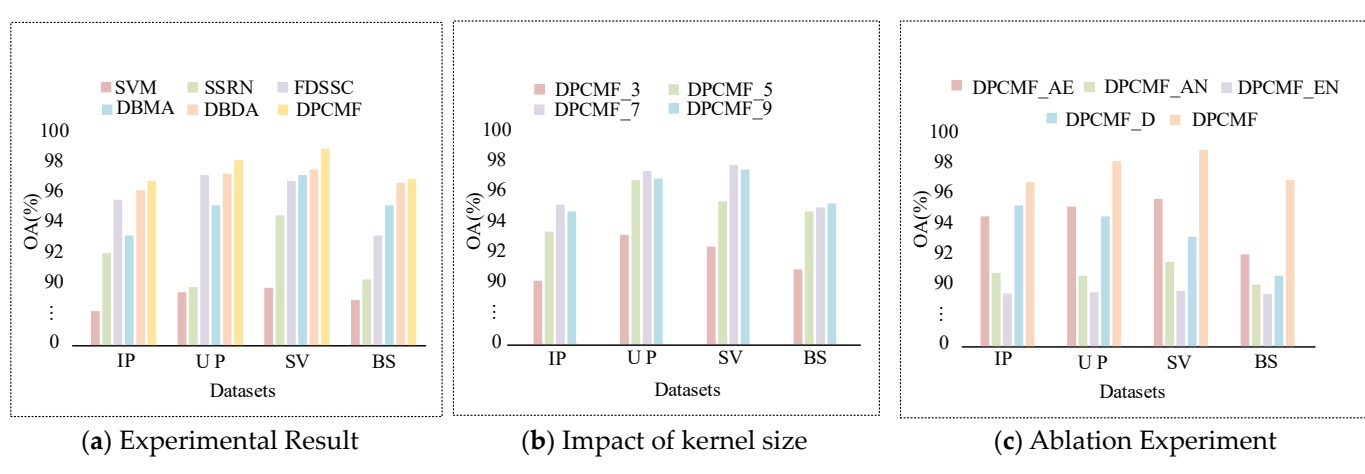

(**a**) Experimental Result  (**b**) Impact of kernel size  (**c**) Ablation Experiment

**Figure 10.** Experiment of DPCMF network on different datasets.

For the IP dataset, the OA of the DPCMF method is 96.74%, which is 27.39%, 4.79%, 1.29%, 3.89% and 0.61% higher than the OA levels of the other five methods, its AA is 96.12%, which is 30.26%, 7.18%, 1.44%, 9.94% and 0.43% higher than the AA levels of the other five methods, and its Kappa coefficient is 96.32%, which is 31.67%, 5.51%, 1.5%, 4.47% and 0.89% higher than the Kappa of the other five methods. The classification accuracy for each category has reached more than 90%. Compared to the other ground objects, the classification accuracy is lower for the Grass-pasture-mowed because there are fewer training samples for the ground objects, and it is difficult to extract a large amount of feature-related information from a small number of samples during model training.

For the UP dataset, the OA of the DPCMF method is 98.10%, which is 15.03%, 9.78%, 1.18%, 2.82% and 0.98% higher than the OA levels of the other five methods, its AA is 98.33%, which is 16.09%, 5.19%, 1.55%, 2.64% and 0.99% higher than the AA levels of the other five methods, and its Kappa coefficient is 97.77%. The classification accuracy for all categories has reached more than 94%. Compared to the alternative ground-truth features, Self-blocking bricks have lower classification accuracy because the category features of the ground-truth features are not obvious, and it is difficult to extract features with elevated discriminative degree.

For the SV dataset, the OA of the DPCMF method is 98.92%, which is 10.83%, 4.5%, 2.4%, 1.97.05% and 1.61% higher than the OA levels of the other five methods, its AA is 98.76%, which is 7.31%, 2.14%, 0.95%, 1.79% and 1.42% higher than the AA levels of the other five methods, and its Kappa coefficient is 98.73%, which is 12.03%, 4.96%, 3.35%, 2.13%

and 0.92% higher than the Kappa coefficient of the other five methods. The classification accuracy for each category has reached more than 91%.

For the BS dataset, the OA of the DCFE method is 96.67%, which is 18.04%, 6.41%, 3.50%, 1.48% and 0.28% higher than the OA levels of the other five methods, its AA is 97.08%, which is 17.51%, 6.26%, 2.63%, 1.36% and 0.58% higher than the AA levels of the other five methods, and its Kappa coefficient is 96.57%, which is 19.70%, 7.10%, 3.98%, 1.78% and 0.48% than the Kappa coefficient of the other five methods. Compared to the other ground features, Floodplain Meadows 2 has lower classification accuracy because there are fewer training samples for this ground feature, the features of this ground feature are more complex, and feature extraction is more difficult during the training of the model.

From the results of the Figure 10b and Table 12, as the size of the convolutional kernel increased, the OA gradually increased, indicating that larger kernels can better capture a wider range of features. However, when the size of the kernel increased to a certain extent, the OA started to decrease because excessively large kernels may capture noise or irrelevant information, thereby reducing the accuracy of the model. In this group of experiments, the best OA can be achieved through pyramidal convolutions. The reason is that the pyramidal convolution module uses multi-scale kernels to capture features of different scales, thereby improving the model generalization ability. Choosing an appropriate combination of convolutional kernels can also improve the accuracy of the model. For example, for certain tasks, the use of smaller kernels may be more suitable because they can better capture local features. Therefore, when choosing the size of the convolutional kernel, it is necessary to ensure a good balance according to the specific task requirements and data characteristics in order to obtain the best experimental results.

In the experiments on the impact of training sample size, the classification accuracy of the SVM, SSRN, FDSSC, DBMA, DBDA and DPCMF methods all improved as the number of training samples increased. Moreover, the performance gap between different models also narrowed with the increase in the number of training samples. The results of these experiments show that, in the case of limited number of training samples, the DPCMF method can better extract multi-class features in the samples by using densely connected pyramidal convolution layers to capture spectral features and multi-scale spatial features and using non-local modules to capture global spatial information. Therefore, it achieved good classification results.

From the results of Figure 10c and Table 17, it can be seen that the absence of any module will reduce the model accuracy. DPCMF-AE performed poorly because of inadequate perception of global spatial features. When processing images, it is not only necessary to understand the characteristics of each pixel in the image, but it is also necessary to understand the global information such as the structure, background, layout and composition of the image. Such global information can help the model better understand the image and improve its performance in image processing. DPCMF-AN performed poorly because spectral images contain multiple continuous spectral bands, and each spectral band corresponds to different spectral information of different wavelengths. Therefore, they have high dimensionality and rich information that can be used to accurately describe the spectral characteristics of objects. The inability to extract spectral features results in the lack of important spectral information, making the model unable to distinguish and classify different objects. DPCMF-EN performed poorly because considering only the spectral information of pixels is often insufficient to provide sufficient information. For example, when classifying vegetation and non-vegetation, the spectral information of vegetation may vary at different positions, making it difficult to distinguish vegetation from non-vegetation using only spectral information. In this case, it is necessary to consider the spatial information of pixels, the positional relationship between pixels in the image, to improve classification accuracy. DPCMF-D performed poorly because the Dense structure plays an important role in feature extraction. The Dense structure can share features between different layers through feature reuse, which can effectively improve the network expressive ability and enable the network to better learn the complex features of

input data. This mechanism can avoid the problem of vanishing gradients that is often encountered in traditional deep networks, thus improving the network feature extraction ability. The parameter sharing between different layers in the Dense structure can greatly reduce the number of parameters that need to be trained in the network, thereby reducing the network complexity. This makes the network more compact and lightweight, helping to avoid overfitting and improve the network generalization ability. Since each layer in the Dense structure can accept inputs from all previous layers, the network can learn the features of input data more quickly. In addition, the Dense structure can use a shallower network structure to achieve the same performance as traditional networks, which can reduce training time and computational cost. Therefore, the Dense structure is essential in the DPCMF network.

## 5. Conclusions

In this paper, we propose a hyperspectral image classification method based on dense pyramidal convolution and multi-feature fusion to address the difficulty in adequately extracting and exploiting the spatial and spectral information of hyperspectral images when the sample size is limited. In this approach, two branches—i.e., spatial and spectral branches—are designed, and in each branch, dense pyramidal convolution layers are used as feature extractors. In the spatial branch, multiple local and global spatial features in image samples are extracted using dense pyramidal convolution and non-local blocks. In the spectral branch, the spectral features in the image samples are extracted by the dense pyramidal convolution module. Finally, the spatial and spectral features are fused through fully connected layers to obtain classification results.

The results of experiments conducted to compare the proposed method with the SVM, SSRN, FDSSC, DBMA and DBDA methods on four public hyperspectral datasets (Indian Pines, Pavia University, Salinas Valley, and Botswana) show that the DPCMF method achieves the best experimental results in terms of OA, AA and Kappa coefficients. In the follow-up study, we will continue to build more efficient classification models to resolve the problem of limited sample size and further improve the current model classification accuracy for hyperspectral images.

**Author Contributions:** Conceptualization, J.Z., H.J., S.S., J.W., P.Z., W.Z. and L.W.; Writing—original draft, L.Z. All authors have read and agreed to the published version of the manuscript.

**Funding:** This work is partially supported by the Shandong Provincial Natural Science Foundation, China under Grant ZR2020MF006 and ZR2022LZH015, partially supported by the Industry-university Research Innovation Foundation of Ministry of Education of China under Grant 2021FNA01001, partially supported by the Major Scientific and Technological Projects of CNPC under Grant ZD2019-183-006, partially supported by the Open Foundation of State Key Laboratory of Integrated Services Networks (Xidian University) under Grant ISN23-09, partially supported by Zhejiang Provincial Natural Science Foundation of China under Grant LZ22F020002.

**Data Availability Statement:** The data presented in this study are available on request from the corresponding author.

**Conflicts of Interest:** The authors declare no conflict of interest.

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
