# Peer review of "Hyperspectral Image Classification Based on Dense Pyramidal Convolution and Multi-Feature Fusion"

_remotesensing, doi:10.3390/rs15122990_

Round 1

Reviewer 1 Report

Please see the attatched file.

I think English should be improved.

Author Response

Dear Editor and Reviewers,
Thank you very much for taking your time to review this manuscript. We sincerely appreciate all your comments and suggestions! We have addressed each of the comments and made a number of changes to the manuscript in response to your precious comments. We also recheck the English of the manuscript. Our response is given in blue font and changes/additions to the manuscript are given in the red text.

Reviewer 2 Report

1. Section 2.3 "Framework of the DPCMF Network" describes that the size of the image samples after PCA is 145*145*100, and the Output Size of the PCA layer in Table 1. Implementation details of the spatial branch attached later is 9*9 *100, it can be explained in the PCA part whether to use the original image for PCA or something else.

2. Figure 4. "Architecture of the non-local block" in Section 2.2 "Non-local block", each Convolution layer can be stacked squares instead of cylinders to represent the convolution layer, which is easier to understand the architecture diagram. 

3.References can be added "C.Yu, R.Han, M.Song, C.Liu, C.Chang, "Feedback

Attention-Based Dense CNN for Hyperspectral Image Classification", in which the architecture is also divided into two branches: spatial (Spatial) and spectral (Spectral), and the concept of DenseNet is also used for feature extraction. The proposed method can also compare the results with it. 

Author Response

(The authors gave the same response as above.)

Round 2

Reviewer 1 Report

We have no more questions.

Reviewer 2 Report

The authors addressed all the issues. 

The authors addressed all the issues.